# Pregnancy Desirability and Motivational Readiness for Postpartum Contraceptive Use: Findings from Population-Based Surveys in Eight Sub-Saharan African Countries

**DOI:** 10.3390/ijerph21010053

**Published:** 2023-12-29

**Authors:** Otobo I. Ujah, Biodun N. Olagbuji, Chukwuemeka E. Ogbu, Innocent A. O. Ujah, Russell S. Kirby

**Affiliations:** 1Chiles Center, College of Public Health, University of South Florida, Tampa, FL 33612, USA; otobo@usf.edu (O.I.U.);; 2Department of Obstetrics and Gynecology, Federal University of Health Sciences, Otukpo 972261, Nigeria; innocentujah@ymail.com; 3Department of Obstetrics and Gynecology, Ekiti State University, Ado-Ekiti 362103, Nigeria

**Keywords:** pregnancy intention, contraception, postpartum, motivational readiness, transtheoretical model, sub-Saharan Africa (SSA), Performance Monitoring and Accountability 2020 (PMA2020)

## Abstract

This study examined the associations between pregnancy intention and motivational readiness for postpartum contraceptive use. Data for this cross-sectional analysis were derived from nationally representative surveys of the Performance Monitoring and Accountability 2020 (PMA2020) project conducted in eight sub-Saharan African countries. Participants included 9488 nonpregnant women of reproductive age (15–49 years) who had given birth in the last 2 years. Weighted multinomial logistic regression analyses were performed to estimate the odds ratios (OR) and their corresponding 95% confidence intervals (CIs) of the associations of motivational readiness for contraceptive adoption categorized as precontemplation, contemplation, and post-action with pregnancy intention. After adjusting for confounding factors, the findings revealed that women in Côte d’Ivoire and Nigeria who had mistimed pregnancies had significantly higher odds of being in the contemplation vs. precontemplation stage compared to those who had intended pregnancies. Similarly, women who had unwanted pregnancies in Ethiopia were also more likely to be in the contemplation stage. Furthermore, significant differences were observed for women in Burkina Faso, Côte d’Ivoire, and Nigeria regarding the association between mistimed pregnancies and being in the post-action stage. For women who had unwanted pregnancies, this association was significant only in Nigeria. Additionally, the odds of being in the contemplation stage, compared to the post-action stage, for women who had unwanted pregnancies were significantly higher in Ethiopia and Nigeria. These results indicate that recent unintended pregnancies in specific sub-Saharan African countries may motivate women to take action to prevent future unintended pregnancies. The findings underscore the importance of tailored and context-specific approaches in family-planning programs based on the stage of motivational readiness.

## 1. Introduction

Although contraceptive use remains a cost-effective strategy for avoiding pregnancy [1], its adoption among women may vary considerably, especially during the postpartum period. The inaccurate use or nonuse of contraceptives and contraceptive failures have been identified as proximal determinants of unintended pregnancies [2,3,4,5,6]. Globally, between 2015 and 2019, approximately 48% of pregnancies were unintended, with sub-Saharan Africa (SSA) bearing a substantial burden and accounting for 91 unintended pregnancies per 1000 women of reproductive age [7]. Unintended pregnancies have well-documented adverse consequences for maternal, child, and family health and well-being [8]. These consequences may vary by the category of unintended pregnancy—mistimed or unwanted, with more severe implications for unwanted pregnancies [9,10,11].

Given the challenges associated with access to safe and legal abortion, especially in many parts of SSA, women not wishing to continue with an unintended pregnancy end up having unsafe abortions [7]. Unsafe abortions represent a leading cause of maternal morbidity and mortality in SSA [12,13,14]. Therefore, preventing unintended pregnancies remains an important public health priority, especially in the context of the sustainable development agenda. Doing so, nevertheless, requires robust and context-specific evidence to guide the family-planning and reproductive health policies and programs.

While most literature on pregnancy-intendedness has mostly focused on unintended pregnancies as a consequence of contraceptive nonuse or failure, only a handful of studies have examined whether indeed a bidirectional relationship exists. In essence, whether the likelihood of adopting contraceptives postpartum is influenced by a previous experience of an unintended pregnancy remains unclear. Evidence suggests that experiencing negative events can impact the cognitive risk perception of individuals and their subsequent willingness to take precautionary actions [15,16,17]. Consequently, the experience of an unintended pregnancy may shape women’s perception of the associated risks, motivating them to take proactive measures, such as contraception use, to prevent future unintended pregnancies.

There has been growing interest in understanding the effect of unintended pregnancies on reproductive and contraceptive behaviors. Observational studies conducted across different contexts have revealed mixed findings [2,18,19]. For example, studies by Fotso et al. [20] and Bakibinga et al. [21] conducted across different contexts in SSA showed women with a history of unintended pregnancy were more likely to use modern contraceptives. Moreover, findings from a recent study among women in Ethiopia revealed that those who experienced mistimed, but not unwanted, pregnancies were significantly more likely to adopt a contraceptive method in the postpartum period compared to those with wanted pregnancies (HR = 1.23, 95% CI = 1.03–1.46) [22]. Surprisingly, these studies do not sufficiently account for women who were not using any method. In other words, limited evidence exists regarding their intention to use (ITU) contraception, an important person-centered measure of demand for family planning (FP) [23]. A more comprehensive investigation, which jointly considers intentions and behavior, could offer relevant insights towards improving postpartum contraceptive adoption following an unintended pregnancy, particularly in contexts characterized by a high unmet need for FP [24].

The ability to effectively avert an unintended pregnancy, whether incident or repeat, depends on the extent of readiness to adopt effective contraceptive methods. Motivational readiness describes a psychological state of willingness to achieve a desired outcome or adopt a specific behavior [25]. From a theoretical standpoint, the transtheoretical stages of change model (SCM) posits that individuals progress through distinct stages of readiness to initiate or adopt specific health behaviors [26]. These stages include precontemplation, contemplation, preparation, action, and maintenance. The SCM has found application in various health behavior studies, including contraceptive use [27,28,29]. In the context of contraceptive adoption, the SCM allows for categorizing women based on their readiness for behavior change, ranging from having no intention to use contraception to consistently maintaining contraceptive use [30,31,32,33]. Furthermore, considering that both behavioral and motivational dimensions are integral to health-related behavior change [34], the SCM provides a valuable framework for understanding the complex dynamics involved in contraceptive utilization. In this paper, we contribute new empirical evidence by using nationally representative data from eight SSA countries to investigate whether and to what extent the intendedness of a previous pregnancy is associated with motivational readiness for contraceptive adoption in the postpartum period.

## 2. Materials and Methods

### 2.1. Study Design and Data 

Data for this population-based cross-sectional analysis were derived from the Performance Monitoring and Accountability 2020 (PMA2020) surveys which are nationally and regionally representative surveys conducted annually or semi-annually across several countries in sub-Saharan Africa and Asia. The surveys employ a multi-stage stratified cluster-sampling technique and utilizes mobile technology to collect data on various reproductive health indicators, including family-planning services, from women (15–49 years), households, and service delivery points (SDPs). In the initial stage, a random selection of enumeration areas (clusters) representative of the entire country was conducted based on the sampling frame for each country. This selection process relied on the latest available national census for each country and utilized a probability proportional to size (PPS). This was followed by the random selection of households from a list of all households in each enumeration area in the second stage. In the final stage, all eligible women aged 15–49 years residing in the selected households were invited to participate in the survey after providing informed consent. Additional details about the survey methodology have been published elsewhere [33]. For this study, we utilized data from the female interviews.

### 2.2. Outcome Variable

The primary outcome was readiness to adopt a contraceptive method in the postpartum period. Eligible female respondents were asked, “*Are you or your partner currently doing something or using any method to delay or avoid getting pregnant?*” Guided by the stages of change model, those who answered “yes” were categorized as being in the post-action stage (i.e., action or maintenance) and were further classified according to the effectiveness of the method they were reportedly using (most effective, moderate, and least effective). Those using more than one method were classified based on the method with higher effectiveness. 

Those who answered “no” were further asked, “*You said that you are not currently using a contraceptive method. Do you think you will use a contraceptive method to delay or avoid getting pregnant at any time in the future?*”. As shown in Figure 1, those who responded with “no” were classified as being in the precontemplation stage (no ITU). Those who answered “yes” were classified as being in the contemplation stage (ITU). Motivational readiness was modeled as a 3-level outcome variable.

### 2.3. Exposure Variable

The primary exposure of interest was pregnancy intention for the most recent pregnancy, which was determined using the conventional timing-based measure of pregnancy intention. During the survey, women were asked “*At the time you became pregnant, did you want to become pregnant then, did you want to wait until later, or did you not want to have any children at all?*”. Women who answered then were classified as having had an intended pregnancy. Those who indicated later and not at all were categorized as having mistimed and unwanted pregnancies, respectively.

### 2.4. Covariates

The plausible factors associated with pregnancy intention and postpartum contraceptive use readiness were defined *a priori* based on previous studies [2,18,19,22], and were included in the analysis if they were measured consistently across the surveys. Maternal age was categorized into three groups: 15–24, 25–34, and 35–49 years. Level of educational attainment was categorized as none, primary, secondary, or higher. Parity, which was measured based on women’s reports of the total number of live births ever had, was categorized as low (1–2), average (3–4), and high (5 or more). Fertility intention was categorized into wants no more, wants more, and ambivalent, and household wealth index was divided into low, middle, or high. The following variables were dichotomized: marital status (not married or in union, or married or in union), place of residence (rural or urban), receipt of family-planning counseling in the community or at a health facility (yes or no) and exposure to family-planning mass media (exposed or not exposed), based on women’s reports of exposure to family-planning messages through at least one media channel, such as radio, television, newspapers, billboards/posters, magazines, or brochures/leaflets).

### 2.5. Analytic Sample

For this analysis, data from recent available cross-sectional and nationally representative PMA survey rounds conducted across eight sub-Saharan African geographies were used and include the follow countries: Burkina Faso (2019), Côte d’Ivoire (2018), Ethiopia (2018), Ghana (2017), Kenya (2018), Niger (2018), Nigeria (2018), and Uganda (2018). Table 1 shows the survey response rates and the number of women interviewed in each survey by country and subregion. This analytic sample was restricted to nonpregnant women of reproductive age (15–49 years) with last childbirth occurring within the 2 years preceding the survey and who provided data on pregnancy intention, and contraceptive use or intentions (for those not using a method of contraception). Further, we excluded women who reported being menopausal, who had a hysterectomy, or who stated they could not become pregnant. The final sample size for each country are as follows: Burkina Faso (*n* = 797), Côte d’Ivoire (*n* = 740), Ethiopia (*n* = 1428), Ghana (*n* = 736), Kenya (*n* = 1057), Niger (*n* = 900), Nigeria (*n* = 2644), and Uganda (*n* = 1186).

### 2.6. Statistical Analysis

We employed univariate, bivariable, and multivariable approaches for data analyses, and separate analyses were conducted for each country. All statistical analyses were performed using SAS version 9.4 (SAS Institute, Inc., Cary, NC, USA) and incorporated the SAS survey procedures to account for the complex sampling design of the PMA2020 surveys to ensure robust estimation of the effect estimates and standard errors (SEs), by including the sample weights, clusters, and strata variables in the analyses. 

For descriptive analysis, we used the SAS SURVEYFREQ procedure to calculate weighted frequencies and percentages (%) for categorical variables, while the SURVEYMEANS procedure was used to generate weighted means and standard errors (SEs) for continuous variables. We conducted bivariable analyses, using the Rao-Scott chi-square test, to examine between-group differences among the categorical variables by the outcome variable. To investigate the association between pregnancy intention and contraceptive use readiness, while adjusting for the identified confounders, we fitted survey-weighted multivariable multinomial logistic regression models using the SURVEYLOGISTIC procedure with the *glogit* function. The results presented are based on a complete-case analysis. We reported effect estimates as both unadjusted and adjusted odds ratios (ORs), accompanied by their corresponding 95% confidence intervals (CIs). All tests were two-tailed, and statistical significance was determined at a *p* value of 0.05.

## 3. Results

### 3.1. Descriptive Statistics

A total of 9488 women from eight different SSA countries were included in the study. Among the surveyed countries, there was substantial heterogeneity in the levels of readiness for contraceptive adoption. Overall, the estimated (unweighted) prevalence of women in the precontemplation, contemplation, and post-action stages was 25.8%, 37.3%, and 36.95, respectively. Among those in the post-action stage, 35% of women are in Burkina Faso, 25% in Côte d’Ivoire, 37% in Ethiopia, 38% in Ghana, 67% in Kenya, 26% in Niger, 27% in Nigeria, and 41% in Uganda (Table 2). The prevalence of women in the precontemplation stage ranged from 11.7% in Kenya to 48.2% in Niger. Similarly, the prevalence of being in the contemplation stage ranged from 21% in Kenya to nearly 50% in in Côte d’Ivoire. Table 2 provides an overview of the weighted percentage distribution of socio-demographic characteristics among women in these eight SSA countries. 

Women’s intendedness of a previous pregnancy also varied by country. Across all countries, at least 55% of women reported their last pregnancy as intended, ranging from 55.1% in Ghana to 90.9% in Niger. Approximately 18%–34% of women across all countries, except Niger, reported their last pregnancy as mistimed; in Niger, nearly 9% reported their last pregnancy as mistimed. In Burkina Faso, Côte d’Ivoire, and Niger, fewer than 3% of women (1.2%, 2.2%, and 0.6%, respectively) reported their last pregnancy as unwanted. In contrast, in all other countries, 6.2%–18.3% of women reported their last pregnancy as unwanted. In the sample, most women (at least 42%) were in the 24–34 age group, while fewer women were in the 35–49 age category. Furthermore, at least 40% of the women in the sample were of low parity (1–2 births) and more than 90% were either currently married or in a union. There were also substantial variations in the educational attainment of women across all countries. Nearly half or more of the women in Burkina Faso, Côte d’Ivoire, Ethiopia, and Niger reported no formal education. The highest levels of primary education were in Uganda (60%), Ethiopia (37%), and Kenya (48%), while the highest levels of secondary education were in Ghana (60%), Nigeria (55%), and Kenya (46%). 

In terms of fertility intention, more than two-thirds of women across all countries expressed a desire for more children. The percentage of women who had no further intention for childbearing were highest in Ethiopia (24%), Ghana (25%), Kenya (32%), and Uganda (28%). In all countries, except Uganda, at least 50% or more of women did not receive family-planning counseling, either in the community or during health facility visits in the past year. A greater proportion of women who received family-planning counseling were in Burkina Faso (49%), Ghana (43%), Kenya (49%), Nigeria (49%), and Uganda (54%). Approximately 50% or more of women across all countries, except Ethiopia, were exposed to at least one form of family-planning mass media. In Ethiopia, 61% of women were not exposed to any form of family-planning mass media. Differences were also observed in the household wealth index across all countries. In most countries, a considerable proportion of women belonged to the lowest wealth category, except for those in Niger. For the highest wealth category, the majority of women were in Ethiopia (36%), Ghana (31%), Kenya (35%), Niger (35%), and Nigeria (32%). Except for Côte d’Ivoire, more than half of women resided in rural areas. About 48% and 53% of women living in Côte d’Ivoire and Nigeria, respectively, resided in urban areas.

### 3.2. Factors Associated with Motivational Readiness for Postpartum Contraceptive Use

Figure 2 shows the distribution of the bivariable relationship between pregnancy intention and levels of motivational readiness for postpartum contraceptive use in eight SSA countries. There was a significant relationship between pregnancy intention and motivational readiness for contraceptive adoption in all countries, except Niger. Across all countries and at each stage of change, women were most likely to report their previous pregnancy as intended, followed by mistimed pregnancies, with the lowest likelihood associated with experiencing an unwanted pregnancy. Overall, among women in the post-action stage, nearly one-half (49.7%) were using a moderately effective method. About 29% were using a most effective method while one-fifth (21.1%) were using a least effective method. These patterns, however, varied substantially across different countries as depicted by the Sankey diagram in Figure 3. The distribution of other demographic and reproductive characteristics, such as age, parity, marital status, education level, fertility intention, receipt of family-planning counseling, exposure to family-planning mass media, household wealth index, and place of residence, also showed substantial variations across stages in multiple countries, with most of these differences being statistically significant (Table 3).

### 3.3. Multinomial Regression Analyses

Table 4 presents the crude and adjusted results of the multinomial logistic regression models for motivational readiness for postpartum contraceptive adoption and pregnancy intentions. In the crude analyses, women across all the SSA countries studied, who reported their pregnancies as mistimed, had higher odds of being in the contemplation stage (reference: precontemplation) compared to those who reported their pregnancies as intended. This relationship was, however, statistically significant for women in Burkina Faso (OR = 2.40, 95% CI = 1.10–5.24, *p* = 0.028), Côte d’Ivoire (OR = 2.15, 95% CI = 1.46–3.16, *p* = 0.0002), Ethiopia (OR = 1.58, 95% CI = 1.00–2.50, *p* = 0.048), Nigeria (OR = 1.93, 95% CI = 1.25–2.97, *p* = 0.0029), and Uganda (OR = 1.75, 95% CI = 1.05–2.91, *p* = 0.032). For women with unwanted pregnancies, the association was less consistent across countries, with statistically significant higher odds observed only in Ethiopia (Crude OR = 3.04, 95% CI = 1.40–6.60, *p* = 0.005), Ghana (OR = 2.09, 95% CI = 1.09–3.97, *p* = 0.025), and Uganda (Crude OR = 2.37, 95% CI = 1.21–4.65, *p* = 0.012). In the multivariable analyses adjusting for potential confounders, the association between mistimed pregnancy and being in the contemplation stage (reference: precontemplation) remained statistically significant for women in Côte d’Ivoire (aOR = 1.95, 95% CI: 1.20–3.18, *p* = 0.0081), Ethiopia (aOR = 2.69, 95% CI = 1.15–6.31, *p* = 0.023), and Nigeria (aOR: 1.97, 95% CI = 1.25–3.17, *p* = 0.0041). For the association between unwanted pregnancy and being in the contemplation stage, statistical significance remained only for women in Ethiopia (aOR = 2.69, 95% CI = 1.15–6.31, *p* = 0.023) after adjusting for potential confounders.

The results of the crude analyses showed that women across all the studied countries, who reported their pregnancies as mistimed, had higher odds of being in the post-action stage (reference: precontemplation) compared to those who reported their pregnancies as intended. However, statistically significant results were observed only for women in Burkina Faso (OR = 2.78, 95% CI = 1.29–5.79, *p* = 0.0094), Côte d’Ivoire (OR = 2.27, 95% CI = 1.36–3.16, *p* = 0.0023), and Nigeria (OR = 2.15, 95% CI = 1.28–3.60, *p* = 0.0037). After adjusting for potential confounders, the associations between mistimed pregnancy and being in the post-action stage remained consistent across these countries, mirroring the findings observed in the unadjusted models. 

Notably, in the crude analyses for unwanted pregnancy, statistically significant associations, with the odds of being in the post-action stage (reference: precontemplation) relative to women who reported pregnancies as intended, were observed in Ghana (OR = 1.91, 95% CI = 1.04–3.52, *p* = 0.039) and Nigeria (OR = 2.15, 95% CI = 1.28–3.60, *p* = 0.0037). However, after adjusting for potential confounders, the association remained statistically significant only for women in Nigeria (aOR = 2.19, 95% CI = 1.19–4.04, *p* = 0.012). For all other countries, there were no statistically significant associations observed in the relationship between having an unwanted pregnancy and being in the post-action stage relative to women who reported pregnancies as intended.

Estimates of the association between pregnancy intention and being in the contemplation (reference: post-action stage) stage showed variations across countries. While the crude association for women with mistimed pregnancy showed significantly higher odds for women in Kenya (OR = 1.61, 95% CI = 1.13–1.21, *p* = 0.0093) and Uganda (OR = 1.55, 95% = 1.12–2.13, *p* = 0.0085), there were no significant differences across all countries after adjusting for confounders. The crude association for women in the unwanted pregnancy category showed significantly higher odds for women in Ethiopia (OR = 2.97, 95% CI = 1.64–5.38, *p* = 0.0004) and Kenya (OR = 2.64, 95% CI = 1.40–4.96, *p* = 0.0029). After adjusting for confounders, statistically significant differences were observed for women in Ethiopia (OR = 2.34, 95% CI = 1.20–4.58, *p* = 0.0125) and Nigeria (OR = 0.45, 95% CI = 0.22–0.94, *p* = 0.0345). 

## 4. Discussion

### 4.1. Main Findings and Comparison with Existing Literature

This study, framed by the transtheoretical stages of change model (SCM), used population-based nationally representative cross-sectional data to investigate the associations between the intendedness of a recent birth with levels of motivational readiness for postpartum contraception in eight SSA countries. We focused our analysis on women who gave birth within the past year or two, based on existing literature suggesting that unmet need is concentrated among women who gave birth within the last year or two [35]. Our results show mixed and varied relationships across the different countries included in this study. After adjusting for multiple confounding factors, we found that the likelihood of being in either the contemplation or post-action states (with precontemplation as the reference) was significantly higher for women in Côte d’Ivoire and Nigeria who had mistimed pregnancies compared to their counterparts whose pregnancies were intended. Additionally, in Burkina Faso, women with mistimed pregnancies were more likely to be in the post-action stage compared to those with intended pregnancies. These results suggest that women with a recent history of mistimed pregnancies in these countries may have recognized their vulnerability and initiated actions to prevent the recurrence of unintended pregnancies. However, women in Kenya, Niger, and Uganda, who experienced a prior unintended pregnancy (mistimed or unwanted), were no more nor less likely, in comparison to their counterparts with intended pregnancies, to report being in either the contemplation or post-action stages of change relative to being in the precontemplation stage. Similarly, in these countries, together with Burkina Faso, Côte d’Ivoire, and Ghana, there was no significant differences in the adjusted associations between being in the contemplation stage and the post-action stage for women who experienced unwanted pregnancies compared to those with intended pregnancies. Moreover, controlling for confounders among women with unwanted pregnancies, the odds of being in the contemplation stage (with precontemplation as the reference) were significantly higher for women in Ethiopia while the odds of being in the post-action stage (with precontemplation as the reference) were significantly higher for women in Nigeria. Interestingly, for the comparison between being in the contemplation relative to the post-action stage, the odds were significantly higher for women in Ethiopia but significantly lower for women in Nigeria. 

Our estimates of women in the contemplation stage are consistent with previous studies. For instance, estimates from a study by Negash et al. [36] revealed that the overall prevalence of intention to use contraception among selected high-fertility countries, including Burkina Faso, Niger, and Nigeria was 38%. The study also revealed a much higher prevalence of ITU contraception in Burkina Faso (59.2%) and Niger (42.3%) compared to our study. Another similar study among fecund sexually active women in developing countries showed a much higher overall and country-specific prevalence of women in the contemplation stage [37]. Differences between our findings and those of previous studies could result from differences in the composition of the analytic samples in these studies. While our study focused on women whose last delivery was in the past two years, other studies included much larger sample sizes of women regardless of the interval between the survey and last childbirth. In addition, these studies did not account for women’s past pregnancy intentions and women who were already using contraception.

Furthermore, our findings, taken together, suggest that, in countries where a positive association exists between pregnancy intentions and motivational readiness for postpartum contraceptive use, experiencing unintended pregnancies, whether mistimed or unwanted, may inform women’s willingness and readiness to adopt contraception in order to avoid future unintended pregnancies. Studies across several SSA countries also show that women with unintended pregnancy had higher odds of being in the post-action stage rather than in the pre-action stage for contraceptive use (i.e., not using a method) [20,21,22], although these studies operationalized the outcome dichotomously (use vs. nonuse). A striking pattern observed in some existing studies, including ours, is that, in certain countries, while mistimed pregnancies were linked to a higher level of readiness for contraceptive adoption (specifically, being in the post-action stage), there were no significant differences in the levels of motivational readiness for contraceptive adoption between women who had experienced an unwanted pregnancy compared to those who reported their last pregnancy as intended. This finding was surprising, given the expectation that women with unwanted pregnancies might have a heightened cognitive risk assessment, which, in theory, should motivate them to take preventive measures against experiencing another unwanted pregnancy. However, why this association occurred mostly among women with mistimed pregnancy relative to those with unwanted pregnancy remains unclear. It is possible that women are more likely to report an unintended pregnancy as mistimed or even intended rather than unwanted (perhaps, *ex post* rationalization). Additionally, the cultural interpretations and acceptability of unintended pregnancy, which differ across contexts, could offer potential explanations regarding these findings [38,39].

### 4.2. Strengths and Limitations

Our study fills an important gap in the existing literature by accounting for the differential psychological effects of pregnancy intention among a sample of women who were not using a method of contraception. Furthermore, this study employed a theoretical approach in conceptualizing, analyzing, and interpreting our results, which is important as there is increasing emphasis on the need for a theory-guided implementation of evidence-based interventions for health behavior change. Additionally, our study investigated the differential effects of mistimed and unwanted pregnancy rather than employing a simple dichotomous classification, which potentially oversimplifies a multidimensional construct and masks the differential effects of unintended pregnancy. Finally, our study used a large, well-defined, nationally representative sample of nonpregnant women across several SSA countries, allowing for generalizability to the population of nonpregnant women in these countries.

The results of our analysis should be interpreted in light of several limitations. First, the ascertainment of pregnancy intentions relied on self-reported data, with women retrospectively recalling their pregnancy desirability. This approach is susceptible to social desirability bias and recall bias, potentially resulting in an underestimation of the true effects of the association between pregnancy intention and motivational readiness for contraceptive use. Similarly, the survey item used to measure contraceptive use is limited in terms of specificity as it encompasses a broad range of methods, from highly effective ones such as intrauterine devices (IUDs) to less effective methods like Fertility Awareness Based Methods (FABMs), which are often underreported. Consequently, the interpretation of terms like “currently” and “prevent or avoid pregnancy” is left to the discretion of the female respondents. Without further probing, this approach may inadvertently exclude those who are indeed using contraception and potentially introduce bias into the estimates and conclusions drawn from this study. Furthermore, while the stages of change model assume movement across each construct in relation to specific time periods, our study’s cross-sectional and secondary nature limited our ability to fully incorporate the temporal aspects of the model. Lastly, as our analyses were limited to nonpregnant women, the results pertaining to being in the precontemplation and contemplation stages of contraceptive readiness may not be readily generalizable to pregnant women regarding their future intentions to use contraception postpartum.

### 4.3. Implications for Public Health

To enhance the effectiveness of stage-based interventions for postpartum family planning, it is important to consider women’s previous pregnancy intentions, along with the influence of social and cultural contexts in future research, policy development, and programs. Given the substantial variations in cultural definitions and acceptability of pregnancy intentions, it is imperative for future research to investigate whether these associations differ in strength and direction when using measures less affected by prevailing cultural and social norms. For instance, alternative perspectives that bypass cultural biases, such as the London Measure of Unplanned Pregnancy (LMUP) and emotional responses to pregnancy as employed in the study by Zimmerman et al. [22], may offer valuable insights. Furthermore, our study identified instances in some countries where women reported very low rates of unwanted pregnancies. It remains uncertain whether these reports were influenced by cultural or religious beliefs. Consequently, qualitative research studies could provide valuable insights into the intricate dynamics of reporting pregnancy intentions and how these reporting patterns shape motivational readiness for contraceptive adoption.

In terms of practical implications, efforts by women’s and reproductive health providers to address the contraceptive and family-planning needs of women who have experienced an unintended pregnancy should be tailored to the stages of readiness for postpartum contraceptive adoption. For example, women in the pre-contemplation stage who have recently experienced an unintended pregnancy can benefit from contraceptive education that includes information on available contraceptive methods, their effectiveness, potential side effects, and accurate usage while also taking into account the prevailing religious and cultural norms within the different contexts. Additionally, providers can engage in discussions with women to understand their threat perception (severity and susceptibility) regarding unintended pregnancy. For women in the contemplation stage, contraceptive counseling could involve discussions exploring women’s concerns and preferences regarding contraceptive use, in addition to identifying barriers related to their self or response efficacy. As for women in the post-action stage, providing ongoing education and counseling on the chosen contraceptive method, including proper usage, potential side effects, and adherence, is essential. Providers can also offer strategies for managing contraceptive side effects. In the case of women using long-acting reversible contraceptive (LARC) methods, providers can offer services related to insertion, removal, and replacement when necessary. Furthermore, providers should ensure that women are aware of the contraceptive’s duration and available options for continuation.

## 5. Conclusions

In this study, we examined the relationship between women’s prior pregnancy intentions and their motivational readiness for contraceptive adoption in diverse SSA contexts. Our results indicate mixed and inconsistent findings among nonpregnant women who had given birth within the past two years, with variations across different pregnancy intention categories and countries. Findings suggest that women with a recent history of mistimed pregnancy in some SSA countries may have recognized their vulnerability and initiated actions to prevent the recurrence of unintended pregnancies. Therefore, contraceptive and reproductive health programs aimed at reducing incident or recurrent unintended pregnancy should tailor interventions based on women’s stage of motivational readiness for contraceptive adoption. In addition, women’s and reproductive health providers need to prioritize ongoing contraceptive and family-planning education for women across all stages of motivational readiness while taking into consideration religious and cultural norms which exist in different settings.

## Figures and Tables

**Figure 1 ijerph-21-00053-f001:**
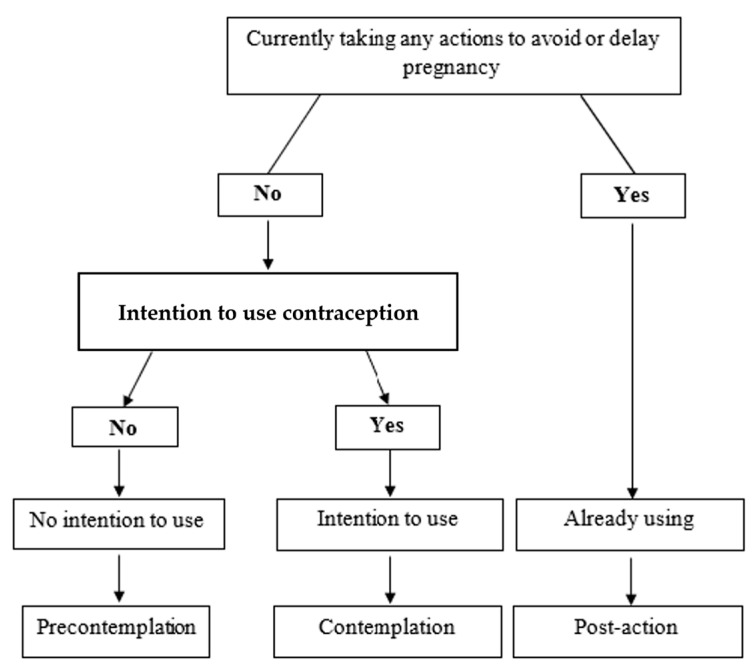
Algorithm to determine stages of motivational readiness for postpartum contraceptive use, adapted from Buchmann et al. [28].

**Figure 2 ijerph-21-00053-f002:**
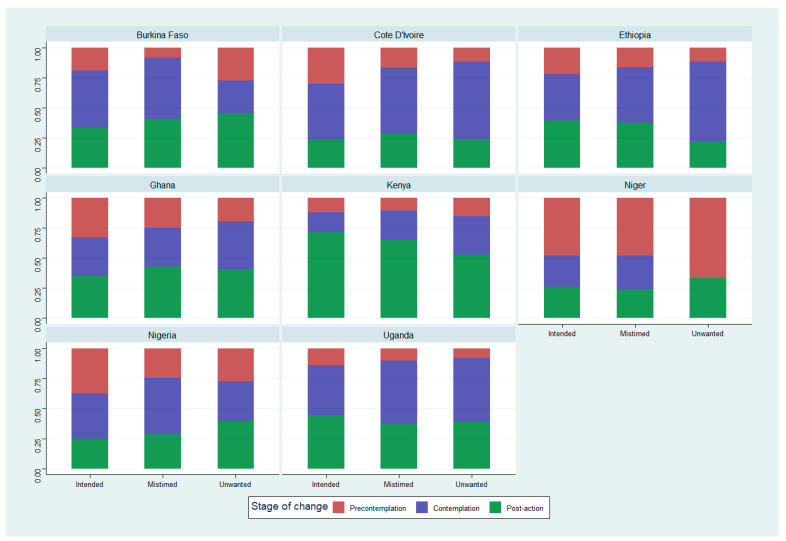
Weighted percentages showing the relationships between maternal pregnancy intention and motivational readiness for contraceptive adoption in eight sub-Saharan African (SSA) countries.

**Figure 3 ijerph-21-00053-f003:**
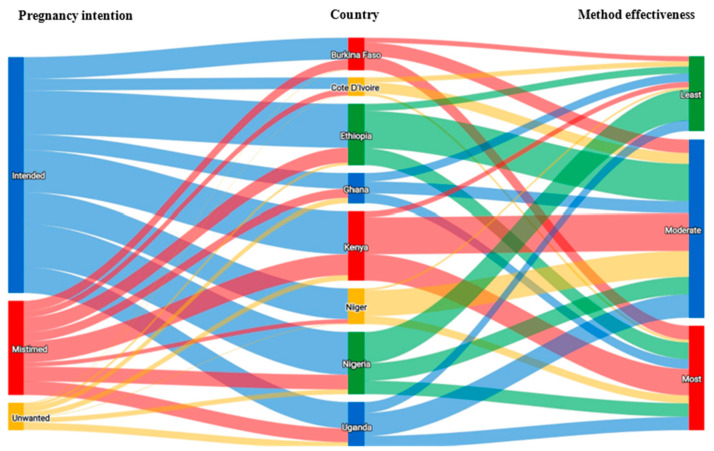
Sankey diagram showing relationships between pregnancy intention, country, and contraceptive method use effectiveness among women in the post-action stage of change.

**Table 1 ijerph-21-00053-t001:** Description of information of PMA2020 survey years, response rates, and analysis sample by country (*N* = 9488).

Country	Survey Year	Response/Completion Rates (%)	Females Interviewed	Sample	Subregion
Ethiopia	2018	98.6	7429	1428	East Africa
Kenya	2018	99.1	5720	1057	East Africa
Uganda	2018	96.8	4225	1186	East Africa
Burkina Faso	2019	97.7	3388	797	West Africa
Côte d’Ivoire	2018	98.1	2738	740	West Africa
Ghana	2017	98.1	4154	736	West Africa
Niger	2017	97.0	3020	900	West Africa
Nigeria	2018	98.1	11,106	2644	West Africa

**Table 2 ijerph-21-00053-t002:** Weighted percentage distribution of socio-demographic characteristics, by countries, of postpartum women in eight sub-Saharan African (SSA) countries, Performance Monitoring Action (PMA) survey, 2017–2018.

Characteristics	Country (Survey Year)							
	Burkina Faso (2019)	Côte d’Ivoire (2018)	Ethiopia (2018)	Ghana (2017)	Kenya (2018)	Niger (2018)	Nigeria (2018)	Uganda (2018)
Number of women (*n*)	886	745	1552	736	1057	1085	2644	1186
All women (%)	100	100	100	100	100	100	100	100
Stage of change of contraceptive readiness								
Precontemplation (no intention to use)	16.70	25.45	19.51	28.23	11.74	48.21	34.53	11.97
Contemplation (intention to use)	48.37	49.59	43.06	33.31	20.86	26.12	38.88	46.92
Post-action (already using)	35.56	24.96	37.43	38.45	67.40	25.68	26.60	41.15
Pregnancy intention								
Intended	69.41	66.02	65.68	55.12	55.74	90.90	75.56	53.78
Mistimed	29.37	31.76	27.09	26.58	33.60	8.99	18.28	32.13
Unwanted	1.22	2.22	7.23	18.31	10.66	0.61	6.16	14.10
Age, years								
15–24	38.81	33.82	29.96	27.75	38.06	38.45	25.50	39.19
24–34	42.79	47.17	51.20	49.76	47.46	42.98	51.35	43.91
35–49	18.4	19.01	18.84	22.49	14.48	18.58	23.16	16.91
Parity								
Low	42.59	44.43	42.84	53.68	56.44	37.22	42.24	40.13
Average	26.25	32.39	26.46	28.20	27.41	27.42	34.47	29.13
High	31.17	32.39	30.70	18.12	16.15	35.36	23.30	30.74
Marital status								
Not married or in union	5.45	12.57	4.33	11.43	17.31	3.32	4.66	14.21
Currently married or in union	94.55	87.43	95.67	88.57	82.69	96.68	95.34	85.79
Highest level of education								
None	68.12	53.31	48.41	20.62	5.98	71.32	27.04	10.08
Primary	15.82	28.79	36.60	19.62	47.74	17.42	18.39	60.01
Secondary or higher	16.06	17.90	14.99	59.76	46.28	11.26	54.57	29.92
Fertility intention								
Want no more	1.89	14.56	23.94	25.35	32.03	3.96	18.01	27.53
Wants more	87.23	82.15	67.15	70.45	62.31	93.39	70.52	69.22
Ambivalent	1.88	3.29	8.46	4.20	5.66	2.65	11.48	3.23
Received family-planning counseling								
No	51.03	65.48	66.56	57.50	50.95	67.16	50.69	46.50
Yes	48.97	34.52	33.44	42.50	49.04	32.84	49.31	53.50
Exposure to family-planning mass media								
Not exposed	36.44	46.15	61.23	24.00	7.28	49.61	31.75	20.62
Exposed	63.56	53.85	38.77	76.01	92.72	50.39	68.25	79.38
Household wealth index								
Lowest	37.85	50.12	43.01	48.42	46.16	29.62	53.64	49.69
Average	37.05	18.99	20.91	20.82	18.64	35.26	14.33	21.14
Highest	25.1	30.89	36.08	30.76	35.20	35.13	32.02	29.17
Place of residence								
Rural	84.14	47.00	78.97	57.68	68.69	83.88	51.66	82.95
Urban	15.86	53.00	21.03	42.32	31.32	16.12	48.34	17.05

Note: All values are weighted percentages to represent the population of women aged 15–49 years.

**Table 3 ijerph-21-00053-t003:** Weighted percentage distributions of characteristics of postpartum women, by stage of readiness to adopt contraception, in Burkina Faso, Côte d’Ivoire, Ethiopia, Ghana, Kenya, Nigeria, Nigeria, and Uganda.

	Burkina Faso				Côte d’Ivoire				Ethiopia			
	Pre-contemplation	Con-templation	Post-action	*p* Value	Pre-contemplation	Con-templation	Post-action	*p* Value	Pre-contemplation	Con-templation	Post-action	*p* Value
Number of women (N)	142	429	315		189	369	186		303	668	581	
All women (%)	100	100	100		100	100	100		100	100	100	
Pregnancy intention												
Intended	82.57	68.47	64.74	**0.037**	78.52	61.97	61.32	**0.01**	73.03	59.77	68.64	**0.002**
Mistimed	15.46	30.83	33.67		20.63	35.04	36.58		22.52	29.17	27.08	
Unwanted	1.96	0.70	1.59		0.85	2.98	2.10		4.44	11.06	4.28	
Age, years												
15–24	38.30	40.86	36.25	0.40	33.29	37.33	27.41	**0.006**	26.38	32.82	28.54	**0.0002**
24–34	38.82	41.16	46.80		44.54	49.62	44.98		43.22	50.54	56.13	
35–49	22.88	17.98	16.95		22.17	13.05	27.62		30.40	16.63	15.34	
Parity												
Low	45.00	41.47	43.02	0.40	41.07	47.16	42.41	0.64	30.60	39.79	52.74	**<0.0001**
Average	19.71	27.07	28.09		31.63	32.27	33.38		24.66	26.37	27.50	
High	35.29	31.47	28.89		27.29	20.56	24.21		44.74	33.84	19.77	
Marital status												
Not married or in union	3.75	5.78	5.76	0.71	9.81	15.10	10.35	0.14	6.17	5.15	2.44	**0.047**
Currently married or in union	96.25	94.22	94.24		90.19	84.90	89.65		93.83	94.85	97.57	
Highest level of education												
None	76.70	71.11	60.18	**0.017**	67.76	53.41	38.36	**<0.0001**	74.70	49.33	33.65	**<0.0001**
Primary	11.67	15.81	17.72		22.12	29.16	34.87		21.00	38.95	42.03	
Secondary or higher	11.64	13.08	22.11		10.12	17.43	26.77		4.30	11.73	24.32	
Fertility intention												
Want no more	10.88	10.95	10.81	0.72	10.28	15.96	16.15	0.27	21.62	28.26	20.18	**0.01**
Wants more	86.51	87.94	86.58		87.41	81.08	78.91		66.03	63.67	72.93	
Ambivalent	2.61	1.11	2.61		2.30	2.96	4.95		12.34	8.07	6.90	
Received family planning counseling												
No	52.07	49.11	53.18	0.63	78.13	63.67	56.19	**0.002**	77.11	66.62	60.99	**0.017**
Yes	47.93	50.89	46.82		21.87	36.33	43.81		22.89	33.38	39.01	
Exposure to family planning mass media												
Not exposed	50.01	32.94	0.031		59.11	48.68	27.90	**<0.0001**	70.44	62.69	54.75	**0.004**
Exposed	49.99	67.06	64.93		40.89	51.32	72.10		29.57	37.31	45.25	
Household wealth index												
Lowest	40.42	42.16	30.82	**0.002**	55.04	60.84	23.81	**<0.0001**	54.38	51.27	27.58	**<0.0001**
Average	34.82	39.75	34.38		20.49	15.83	23.75		24.13	23.44	16.32	
Highest	24.76	18.09	34.80		24.47	23.34	52.44		21.49	25.29	56.10	
Place of residence												
Rural	86.71	88.92	76.48	**<0.0001**	55.36	52.00	28.55	**0.001**	88.47	85.77	66.21	**<0.0001**
Urban	13.29	11.08	23.52		44.64	48.00	71.45		11.53	14.23	33.79	
	**Ghana**				**Kenya**				**Niger**			
	**Pre-contemplation**	**Con-templation**	**Post- action**	***p* Value**	**Pre-contemplation**	**Con-templation**	**Post-Action**	***p* Value**	**Pre-contemplation**	**Con-templation**	**Post-action**	***p* Value**
Number of women (N)	208	246	283		123	249	709		523	283	279	
All women (%)	100	100	100		100	100	100		100	100	100	
Pregnancy intention												
Intended	63.73	52.66	50.93	**0.03**	56.65	44.26	59.13	**0.006**	90.28	90.17	90.85	0.87
Mistimed	23.60	25.46	29.73		29.95	39.22	32.50		8.98	9.74	8.28	
Unwanted	12.68	21.88	19.34		13.40	16.52	8.37		0.75	0.10	0.88	
Age, years												
15–24	23.03	28.08	30.93	0.14	33.12	47.48	36.00	**<0.0001**	39.18	35.64	39.93	0.88
24–34	49.46	52.27	47.82		38.43	41.03	51.02		41.25	46.50	42.64	
35–49	27.51	19.65	21.25		28.45	11.49	12.98		19.57	17.86	17.43	
Parity												
Low	57.51	53.11	51.37	0.73	44.95	61.90	56.74	**<0.0001**	33.99	36.19	44.35	0.33
Average	27.15	28.48	28.47		15.58	21.23	31.38		29.46	25.33	25.69	
High	15.34	18.11	20.16		39.47	16.86	11.87		36.55	38.48	29.96	
Marital status												
Not married or in union	13.09	16.24	6.03	**0.003**	23.99	29.74	12.30	**<0.0001**	4.10	3.44	1.75	0.49
Currently married or in union	86.91	83.76	93.97		76.00	70.26	87.70		95.90	96.56	98.25	
Highest level of education												
None	23.25	20.26	19.00	0.23	26.62	8.23	1.69	**<0.0001**	79.39	66.76	60.81	**<0.0001**
Primary	15.30	23.57	19.37		56.74	54.53	44.08		14.34	22.07	18.46	
Secondary or higher	61.45	56.17	61.63		16.64	37.24	54.23		6.27	11.17	20.73	
Fertility intention												
Want no more	25.43	23.84	26.60	0.15	28.20	32.96	32.41	0.936	3.44	4.63	4.28	0.73
Wants more	67.65	74.01	69.42		66.65	61.00	62.14		94.08	91.52	94.01	
Undecided	6.93	2.14	3.99		6.15	6.03	5.46		2.49	3.85	1.74	
Received family planning counseling												
No	69.72	56.70	49.23	**0.0003**	60.62	47.54	50.32	0.203	73.29	66.7325	56.09	**0.0019**
Yes	30.28	43.31	50.77		39.38	52.46	49.68		26.71	33.2675	43.92	
Exposure to family planning mass media												
Not exposed	25.36	29.51	18.22	**0.017**	21,04	9.20	4.29	**<0.0001**	50.68	45.86	51.41	0.72
Exposed	74.64	70.49	81.78		78.96	90.80	95.71		49.32	54.14	48.59	
Household wealth index												
Lowest	42.56	57.50	44.86	**0.018**	65.58	47.44	42.39	**0.0013**	35.3048	32.43	16.09	**<0.0001**
Average	25.78	18.44	19.23		15.13	21.81	18.27		40.1511	29.85	31.57	
Highest	31.67	24.05	35.91		19.27	30.76	39.34		24.5441	37.73	52.35	
Place of residence												
Rural	47.18	66.82	57.47	**0.011**	85.26	68.39	65.89	**0.0004**	90.4424	86.71	68.70	**<0.0001**
Urban	52.82	33.18	42.53		14.74	31.61	34.11		9.5576	13.30	31.30	
	**Nigeria**						**Uganda**					
	**Pre-contemplation**	**Contemplation**		**Post-action**	***p* Value**		**Pre-contemplation**	**Contemplation**		**Post-action**	***p* Value**	
Number of women (N)	862	973		664			142	560		492		
All women (%)	100	100		100			100	100		100		
Pregnancy intention												
Intended	82.15	72.66		71.23	**0.0009**		63.34	47.60		58.05	**0.013**	
Mistimed	12.94	22.09		19.64			27.64	36.33		28.65		
Unwanted	4.90	5.25		9.13			9.02	16.08		13.31		
Age, years												
15–24	32.75	25.38		16.26	**<0.0001**		38.56	37.72		41.05	0.18	
24–34	44.47	52.59		58.46			39.01	43.62		45.65		
35–49	22.78	22.03		25.28			22.43	18.66		13.30		
Parity												
Low	38.70	44.66		43.27	**0.0064**		39.37	36.91		44.03	**0.014**	
Average	33.51	32.22		38.98			31.59	27.14		30.68		
High	27.79	23.11		17.75			29.04	35.95		25.30		
Marital status												
Not married or in union	5.21	4.97		3.49	0.398		17.52	16.66		10.45	**0.035**	
Currently married or in union	94.79	95.03		96.51			82.48	83.34		89.55		
Highest level of education												
None	48.07	21.43		7.95	**<0.0001**		21.12	11.36		5.41	**<0.0001**	
Primary	20.20	22.33		10.30			53.33	64.33		57.01		
Secondary or higher	31.74	56.25		81.76			25.55	24.31		37.58		
Fertility intention												
Want no more	9.77	19.24		26.90			19.18	31.67		25.24	**0.0098**	
Wants more	73.45	71.11		65.85			74.34	64.86		72.74		
Undecided	16.78	9.66		7.25			19.18	31.67		25.24		
Received family planning counseling												
No	66.72	48.83		32.61	**<0.0001**		55.38	42.50		48.47	**0.027**	
Yes	33.28	51.17		67.39			44.62	57.50		51.53		
Exposure to family planning mass media												
Not exposed	45.34	30.44		16.03	**<0.0001**		21.77	19.70		21.34	0.74	
Exposed	54.66	69.56		83.97			78.23	80.30		78.66		
Household wealth index												
Lowest	72.14	56.02		26.17	**<0.0001**		57.81	57.94		37.93	**<0.0001**	
Average	11.83	14.32		17.61			18.94	18.77		24.49		
Highest	16.04	29.66		56.22			23.25	23.30		37.58		
Place of residence												
Rural	65.73	52.90		31.58	**<0.0001**		84.00	83.62		81.88	0.82	
Urban	34.27	47.10		68.42			16.00	16.38		18.12		

Boldface indicates statistical significance.

**Table 4 ijerph-21-00053-t004:** Crude and adjusted odds ratios (ORs) for pregnancy intentions associated with motivational readiness for contraceptive adoption among women in eight SSA countries.

Country		Contemplation vs. Precontemplation						Post-action vs. Precontemplation					
	Pregnancy Intentions	Crude OR	95% CI	*p* Value	aOR	95% CI	*p* Value	Crude OR	95% CI	*p* Value	aOR	95% CI	*p* Value
Burkina Faso													
	Intended	1.00	Referent		1.00	Referent		1.00	Referent		1.00	Referent	
	Mistimed	**2.40**	**1.10–5.24**	**0.028**	2.12	0.92–4.91	0.078	**2.78**	**1.29–5.79**	**0.0094**	**2.88**	**1.29–6.44**	**0.011**
	Unwanted	0.43	0.08–2.46	0.34	0.59	0.09–3.99	0.58	1.03	0.18–5.83	0.98	1.36	0.22–8.56	0.74
Côte d’Ivoire													
	Intended	1.00	Referent		1.00	Referent		1.00	Referent		1.00	Referent	
	Mistimed	**2.15**	**1.46–3.16**	**0.0002**	**1.95**	**1.20–3.18**	**0.0081**	**2.27**	**1.36–3.80**	**0.0023**	**2.57**	**1.34–4.92**	**0.0052**
	Unwanted	4.46	0.69–28.70	0.11	4.94	0.65–37.23	0.12	3.16	0.37–27.08	0.29	7.97	0.48–13.15	0.15
Ethiopia													
	Intended	1.00	Referent		1.00	Referent		1.00	Referent		1.00	Referent	
	Mistimed	**1.58**	**1.00–2.50**	**0.048**	1.58	0.69–2.59	0.07	1.28	0.77–2.13	0.34	1.41	0.83–2.41	0.21
	Unwanted	**3.04**	**1.40–6.60**	**0.005**	**2.69**	**1.15–6.31**	**0.023**	1.03	0.39–2.72	0.96	1.15	0.42–3.17	0.79
Ghana													
	Intended	1.00	Referent		1.00	Referent		1.00	Referent		1.00	Referent	
	Mistimed	1.31	0.80–2.13	0.28	1.19	0.65–2.19	0.56	1.58	0.96–2.60	0.074	1.57	0.87–2.81	0.13
	Unwanted	**2.09**	**1.09–3.97**	**0.025**	1.69	0.85–3.38	0.13	**1.91**	**1.04–3.52**	**0.039**	1.79	0.89–3.59	0.09
Kenya													
	Intended	1.00	Referent		1.00	Referent		1.00	Referent		1.00	Referent	
	Mistimed	1.68	0.94–2.99	0.08	1.74	0.90–3.38	0.10	1.04	0.59–1.82	0.89	1.31	0.72–2.39	0.38
	Unwanted	1.58	0.72–3.47	0.26	1.49	0.55–4.07	0.43	0.59	0.28–1.27	0.18	0.77	0.32–1.88	0.57
Niger													
	Intended	1.00	Referent		1.00	Referent		1.00	Referent		1.00	Referent	
	Mistimed	1.09	0.45–2.61	0.85	1.07	0.42–2.77	0.88	0.92	0.47–1.80	0.80	1.03	0.49–2.15	0.93
	Unwanted	0.13	0.01–1.69	0.12	0.08	0.01–1.12	0.06	1.17	0.21–6.51	0.86	0.58	0.13–2.49	0.46
Nigeria													
	Intended	1.00	Referent		1.00	Referent		1.00	Referent		1.00	Referent	
	Mistimed	**1.93**	**1.25–2.97**	**0.0029**	**1.97**	**1.25–3.17**	**0.0041**	**1.75**	**1.10–2.79**	**0.019**	**2.05**	**1.25–3.38**	**0.0049**
	Unwanted	1.21	0.69–2.10	0.49	0.99	0.58–1.69	0.96	**2.15**	**1.28–3.60**	**0.0037**	**2.19**	**1.19–4.04**	**0.012**
Uganda													
	Intended	1.00	Referent		1.00	Referent		1.00	Referent		1.00	Referent	
	Mistimed	**1.75**	**1.05–2.91**	**0.032**	1.62	0.97–2.70	0.07	1.13	0.62–2.07	0.67	1.32	0.74–2.34	0.35
	Unwanted	**2.37**	**1.21–4.65**	**0.012**	1.94	0.88–4.27	0.10	1.61	0.76–3.41	0.21	1.89	0.74–4.81	0.18
				**Contemplation vs. Post-action**									
**Country**		**Pregnancy Intentions**		**Crude OR**		**95% CI**		***p* Value**		**aOR**	**95% CI**		***p* Value**
Burkina Faso													
		Intended		1.00		Referent				1.00	Referent		
		Mistimed		0.87		0.53–1.42		0.56		0.74	0.44–1.25		0.26
		Unwanted		0.42		0.08–2.28		0.31		0.43	0.07–2.68		0.36
Côte d’Ivoire													
		Intended		1.00		Referent				1.00	Referent		
		Mistimed		0.95		0.64–1.40		0.79		0.76	0.47–1.22		0.26
		Unwanted		1.41		0.44–4.52		0.56		0.62	0.11–3.66		0.59
Ethiopia													
		Intended		1.00		Referent				1.00	Referent		
		Mistimed		1.24		0.81–1.89		0.33		1.12	0.72–1.74		0.61
		Unwanted		**2.97**		**1.64–5.38**		**0.0004**		**2.34**	**1.20–4.58**		**0.0125**
Ghana													
		Intended		1.00		Referent				1.00	Referent		
		Mistimed		0.83		0.56–1.22		0.34		0.76	0.48–1.20		0.24
		Unwanted		1.09		0.73–1.64		0.66		0.95	0.56–1.60		0.84
Kenya													
		Intended		1.00		Referent				1.00	Referent		
		Mistimed		**1.61**		**1.13–2.31**		**0.0092**		1.33	0.90–1.97		0.15
		Unwanted		**2.64**		**1.40–4.96**		**0.0029**		1.93	0.92–4.07		0.08
Niger													
		Intended		1.00		Referent				1.00	Referent		
		Mistimed		1.19		0.44–3.21		0.74		1.04	0.37–2.94		0.94
		Unwanted		0.11		0.01–1.08		0.06		0.14	0.01–1.83		0.13
Nigeria													
		Intended		1.00		Referent				1.00	Referent		
		Mistimed		1.10		0.78–1.56		0.58		0.97	0.68–1.37		0.85
		Unwanted		0.56		0.28–1.12		0.101		**0.45**	**0.22–0.94**		**0.034**
Uganda													
		Intended		1.00		Referent				1.00	Referent		
		Mistimed		**1.55**		**1.12–2.13**		**0.0085**		1.23	0.91–1.65		0.17
		Unwanted		1.47		0.93–2.32		0.09		1.03	0.62–1.69		0.92

Abbreviations: aOR, adjusted odds ratio; CI, confidence interval; MOD, mode of delivery, OR, odds ratio. Models were adjusted for age, parity, marital status, education, fertility preferences, receipt of family-planning counselling, exposure to family-planning mass media, household wealth index, and place of residence. Bolded indicate statistically significant estimates at *p* < 0.05.

## Data Availability

The anonymized data that support the findings of this study are publicly available and can be accessed from https://www.pmadata.org (accessed on 24 October 2023).

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
