# Peer review of "Pregnancy Desirability and Motivational Readiness for Postpartum Contraceptive Use: Findings from Population-Based Surveys in Eight Sub-Saharan African Countries"

_ijerph, 2023, doi:10.3390/ijerph21010053_

Round 1

Reviewer 1 Report

Comments and Suggestions for Authors

I am impressed with this capably conducted study. I pose the following friendly comments to sharpen some of the arguments.

1. Please provide survey response rates or completion rates.

2. Please note the limitations of the outcome variable. The survey item, as posed, could include either effective or ineffective methods (e.g., withdrawal in the case of the latter). So, the lack of specificity (which the authors cannot rectify) is a limitation. Respondent interpretations of pregnancy prevention actions could be shaped by their perceptions (accurate or inaccurate) on this issue. Note the subjectivity of this item. 

3. Not all covariates are completely clear. Provide additional information on parity.

4. Some typos are evident. For example, on page 17, go with "gave birth *within* the last year or two."

Generally well done!

Comments on the Quality of English Language

Minor spellchecking is recommended. Otherwise, writing is sound.

Author Response

I am impressed with this capably conducted study. I pose the following friendly comments to sharpen some of the arguments.

Many thanks for the kind words and the thoughtful review of this study. We have incorporated all feedback and the revised manuscript has greatly improved as a result.

1. Please provide survey response rates or completion rates.

We have provided this information as Table 1 in the revised manuscript. (See revised manuscript page 5)

2. Please note the limitations of the outcome variable. The survey item, as posed, could include either effective or ineffective methods (e.g., withdrawal in the case of the latter). So, the lack of specificity (which the authors cannot rectify) is a limitation. Respondent interpretations of pregnancy prevention actions could be shaped by their perceptions (accurate or inaccurate) on this issue. Note the subjectivity of this item

Thank you for this thoughtful suggestion.

We have included this in the revised manuscript where we state;

“Similarly, the survey item used to measure contraceptive use lacks specificity and encompasses a broad range of methods, from highly effective ones such as intrauterine devices (IUDs) to less effective methods like Fertility Awareness Based Methods (FABMs), which are often underreported. Consequently, the interpretation of terms like “currently” and “prevent or avoid pregnancy” is left to the discretion of the female respondents. Without further probing, this approach may inadvertently exclude those who are indeed using contraception and potentially introduce bias into the estimates and conclusions drawn from this study.” (See revised manuscript page 22)

3. Not all covariates are completely clear. Provide additional information on parity.

We have provided additional information to provide further clarification regarding parity.

Parity was defined based on women’s reports of the total number of live births ever had and classified as low (1-2), average (3-4) and high (5+) (See revised manuscript page 3)

4. Some typos are evident. For example, on page 17, go with "gave birth *within* the last year or two."

Thank you for this suggestion. We have noted and effected this change.

Reviewer 2 Report

Comments and Suggestions for Authors

Review:

Dear authors, it is a very good manuscript.

I would like to ask what does this manuscript add to the subject area compared with other published material?

I would like to know your opinion about why there is no differences in the level of motivational readiness for contraceptive method between women who had experienced an unwanted pregnancy compared to those who reported their last pregnancy as intended. (line 408-410). Do you think the questionnaire should be modified? How would you like to improve the methodology?

It would be useful an explanation regarding the opinion of the authors regarding why the motivation to use contraceptive measures occurred mostly among women with mistimed pregnancy relative those with unwanted pregnancy. (line 428-429)In this regard , do the author think if this results answer the main question of the study?

I like the the way that the authors highlighted the practical implications. ( line461-478).

Figure 3 with Sankey diagram is interesting and clever.

I believe that the discussions and conclusions should contain solutions related to education regarding contraception and family planning, taking into account their cultural and religious customs.

Author Response

Dear authors, it is a very good manuscript.

Many thanks for the kind words and the thoughtful review of this study. We have incorporated all feedback and the revised manuscript has greatly improved as a result.

I would like to ask what does this manuscript add to the subject area compared with other published material?

First, although there has been a growing interest among scholars to understand the impact of unintended pregnancy on contraceptive and reproductive outcomes, only a handful of studies are available. Even among existing studies, there is an explicit lack of research in sub-Saharan Africa (SSA), where the burden of unintended pregnancy and maternal mortality is high. Existing studies in SSA are mainly conducted in East Africa, with none in West Africa. Therefore, this study contributes to the body of knowledge regarding the impact of unintended pregnancy on subsequent contraceptive behavior dynamics in the subregion.

Secondly, most studies on this topic have focused only on contraceptive use (yes or no) and by contraceptive method, without relying on theoretical foundations or even appropriately accounting for those who are not using a method of contraception. Importantly, health behavior theories provide a framework for identifying and explaining the mechanisms between various factors associated with health behaviors. By using the Stages of Change Model, we are able to understand where exactly women are in terms of their decisions or intentions to use contraception. The novelty of this study lies in the application of a health behavior theory, an approach not considered in other published material. This understanding is crucial as it can help tailor interventions such as counseling and other contraceptive service delivery interventions to women based on their stage of change.

Thirdly, we examine the patterns of motivational readiness for contraceptive use in the postpartum period across different sub-Saharan African contexts. This provides nuanced insights into how contraceptive attitudes and behaviors following unintended pregnancy vary across different contexts and highlights the factors that may play a key role in these behaviors.

I would like to know your opinion about why there is no differences in the level of motivational readiness for contraceptive method between women who had experienced an unwanted pregnancy compared to those who reported their last pregnancy as intended. (line 408-410). Do you think the questionnaire should be modified? How would you like to improve the methodology?

In our opinion, we believe that the lack of a clear association between levels of motivational readiness for women with unwanted pregnancy compared to those whose pregnancy was intended could be due to conceptual and methodological factors.

Conceptually, one consideration could be how women define and interpret unwanted pregnancy. It is possible that different women have different interpretations of what constitutes an unwanted pregnancy, which may affect their motivation levels.

Furthermore, there are multiple dimensions of pregnancy intention, including cognitive, affective, and contextual dimensions. However, in our study, we only assessed the cognitive dimension using the conventional timing-based approach. It is plausible that there may be an association between the other dimensions of pregnancy intention that were not captured in our study.

Methodologically, the lack of association could be a result of residual confounding. Due to the limitations of the data used in our analysis, we were unable to adjust for additional confounding factors because data on these factors were not collected. This methodological limitation could potentially impact the estimates from our analysis and, consequently, the conclusions drawn from it.

While we do not believe that the questionnaire used in our study should be modified, we do think that future research on this topic should consider alternative approaches to measuring pregnancy intention. For example, the London Measure of Unplanned Pregnancy (LMUP) or the Emotional Response to Pregnancy approach could provide different perspectives and better capture the complexity of pregnancy intention. Taking into account these different approaches may help to address the ongoing debates about the nature of pregnancy intentions and ultimately lead to a better understanding of this topic

It would be useful an explanation regarding the opinion of the authors regarding why the motivation to use contraceptive measures occurred mostly among women with mistimed pregnancy relative those with unwanted pregnancy. (line 428-429) In this regard , do the author think if this results answer the main question of the study?

We have provided an explanation for these findings in the revised manuscript (See page 23, line 396-410)

We believe that the results adequately address the main research question of our study (i.e., whether the intendedness of a recent birth is associated with levels of motivational readiness for postpartum contraception in eight SSA countries).

I like the the way that the authors highlighted the practical implications. ( line461-478).

Many thanks for the compliment

Figure 3 with Sankey diagram is interesting and clever.

Many thanks for the compliment

I believe that the discussions and conclusions should contain solutions related to education regarding contraception and family planning, taking into account their cultural and religious customs

We have highlighted the role of contraceptive and family planning education as part of the practical implications and recommendations based on the results of our study in the revised manuscript (See page 24, lines 459-473) and in the conclusion (See page 24, lines 488-491).

Reviewer 3 Report

Comments and Suggestions for Authors

1.       This is a good manuscript. Introduction is well written, methods & material section and conclusion are fine.

2.       But the results and discussion could be better presented. Do not lump all the findings together. I suggest you break the results/findings into sub-sections, each sub-section with a heading – so that the reader will know what is being presented under each sub-section.  It will also make it easier for the reader to follow what is being presented. You may follow this example or outline:

                    3.1 Socio-demographics characteristics. ( Present socio demographics. Focus should be on

                        table 1)

                    3.2. Women’s pregnancy & fertility intentions…(focus on table 1.)

                    3.3. Pregnancy & Motivational readiness… (focus on figure 2, 

                       bivariable analyses, table 2,  etc.)

                    3.4 Pregnancy & Motivational readiness (multinominal logistic

                      regression/multivariable analyses, table 2…

3.       I think the discussion is lengthy, try to shorten your discussion section. Please do not discuss everything.  Discuss the key findings and support those key findings with previous studies before you discuss the limitations and strengths. Also, consider breaking the discussion section into two: discussion and implications.  You may also follow this outline.

 4.1. Discussion (first discuss the key findings while you support key findings with other studies, and then add limitations and strengths)

3.2    Implications (present the study implications). 

4.       Why are you bringing the extended parallel process model (EPPM) in your discussion?  (Refer to page 18, lines 423-424) Yes, there may be a need to build women’s contraceptive use self-efficacy. But the transtheoretical model also addresses self-efficacy, so why the EPPM?

Comments on the Quality of English Language

English language is fine but may need minor editing.

Author Response

1.       This is a good manuscript. Introduction is well written, methods & material section and conclusion are fine.

Many thanks for the kind words and the thoughtful review of this study. We have incorporated all feedback and the revised manuscript has greatly improved as a result.

2.       But the results and discussion could be better presented. Do not lump all the findings together. I suggest you break the results/findings into sub-sections, each sub-section with a heading – so that the reader will know what is being presented under each sub-section.  It will also make it easier for the reader to follow what is being presented. You may follow this example or outline:

                    3.1 Socio-demographics characteristics. ( Present socio demographics. Focus should be on

                        table 1)

                    3.2. Women’s pregnancy & fertility intentions…(focus on table 1.)

                    3.3. Pregnancy & Motivational readiness… (focus on figure 2, 

                       bivariable analyses, table 2,  etc.)

                    3.4 Pregnancy & Motivational readiness (multinominal logistic

                      regression/multivariable analyses, table 2…

Thank you for this suggestion. We have presented the results under 3 different subsections (descriptive statistics, factors associated with motivational readiness for postpartum contraception and multinomial regression analyses). By doing so, we believe that readers will find it easier to follow and understand the results presented.

3.       I think the discussion is lengthy, try to shorten your discussion section. Please do not discuss everything.  Discuss the key findings and support those key findings with previous studies before you discuss the limitations and strengths. Also, consider breaking the discussion section into two: discussion and implications.  You may also follow this outline.

 4.1. Discussion (first discuss the key findings while you support key findings with other studies, and then add limitations and strengths)

3.2    Implications (present the study implications). 

We have shortened the discussion and organized it into 2 major subsections to enhance clarity and readability.

4.       Why are you bringing the extended parallel process model (EPPM) in your discussion?  (Refer to page 18, lines 423-424) Yes, there may be a need to build women’s contraceptive use self-efficacy. But the transtheoretical model also addresses self-efficacy, so why the EPPM?

We have now revised the manuscript without including the EPPM in order to avoid repetition and any confusion this may create.

Round 2

Reviewer 3 Report

Comments and Suggestions for Authors

Authors have revised the paper based my comments. It looks good.

Comments on the Quality of English Language

It may need minor editing.